# Machine Learning-Based Simulation of the Air Conditioner Operating Time in Concrete Structures with Bayesian Thresholding

**DOI:** 10.3390/ma17092108

**Published:** 2024-04-29

**Authors:** Changhwan Jang, Hong-Gi Kim, Byeong-Hun Woo

**Affiliations:** 1Department of Smart Construction and Environmental Engineering, Daejin University, 1007 Hoguk-ro, Pocheon-si 11159, Republic of Korea; cjang@daejin.ac.kr; 2Civil and Environmental Engineering Department, Hanyang University, Jaesung Civil Engineering Building, 222 Wangsimni-ro, Seongdong-gu, Seoul 04763, Republic of Korea

**Keywords:** Bayesian, threshold, smRNN, concrete, power consumption

## Abstract

Efficient energy use is crucial for achieving carbon neutrality and reduction. As part of these efforts, research is being carried out to apply a phase change material (PCM) to a concrete structure together with an aggregate. In this study, an energy consumption simulation was performed using data from concrete mock-up structures. To perform the simulation, the threshold investigation was performed through the Bayesian approach. Furthermore, the spiking part of the spiking neural network was modularized and integrated into a recurrent neural network (RNN) to find accurate energy consumption. From the training-test results of the trained neural network, it was possible to predict data with an R^2^ value of 0.95 or higher through data prediction with high accuracy for the RNN. In addition, the spiked parts were obtained; it was found that PCM-containing concrete could consume 32% less energy than normal concrete. This result suggests that the use of PCM can be a key to reducing the energy consumption of concrete structures. Furthermore, the approach of this study is considered to be easily applicable in energy-related institutions and the like for predicting energy consumption during the summer.

## 1. Introduction

Efficient energy consumption has become increasingly important due to environmental challenges in recent years. Carbon emissions make this planet hotter every year, and this phenomenon is directly related to the use of energy. Hot weather makes people use air conditioners to cool their rooms, which, in turn, heats the outside air. This vicious cycle has repeated over the years, contributing to the current abnormal climate conditions. Climatologists have long warned of these climate changes and vicious cycles. The cement/concrete industry, which accounts for a large portion of carbon emissions, is making various efforts to take responsibility for the abnormal climate. For example, carbon generation can be limited by reducing the amount of cement used. Chalee et al. [1] used high-volume fly ash as a replacement material for cement and the replacement ratio was up to 60% of the cement weight. Saeed et al. [2] replaced the cement with slag up to 45%. Kim et al. [3] replaced the cement with biomass wood fly ash up to 30%. These studies demonstrated that, while maintaining mechanical properties comparable to ordinary cement, durability generally improved. In a similar but improved way, studies on cement-free geopolymers are being undertaken as well. Ates et al. [4] tried to improve the performance of fly ash-based geopolymer by replacing biomass wood fly ash with up to 50% of fly ash in some parts. Park et al. [5] investigated the volume expansion of a metakaolin-based geopolymer when adding silica fume. Zhang et al. [6] studied the behavior changes of a fly ash-slag-based geopolymer when adding silica fume. The key commonality of these by-product-based geopolymer studies is that cement usage is 0%. In addition to the introduced cases, there are many other studies that have investigated reducing the use of cement or using geopolymer [7,8]. Efforts to reduce cement use are considered an important method to decrease carbon emissions.

Another strategy for addressing climate issues involves the use of non-natural aggregates. Non-natural aggregates include recycled aggregates. Wang et al. [9] investigated the effect of use of coarse/fine slag aggregate on the mechanical properties of concrete. The results indicated that the mechanical properties were similar to those of ordinary concrete, although workability was significantly reduced. Li [10] studied the mechanical properties of concrete using limestone powder-treated recycled aggregate. Treating aggregate had the effect of increasing the mechanical properties of the concrete. In addition, the study of Li [10] estimated the value of using recycled material in the construction field. There are many studies on the use of recycled materials [11,12,13,14] like the studies of Wang et al. [9] and Li [10]. Some studies have also focused on the function of materials used in aggregates. Studies examining thermal properties have addressed issues such as enhancing thermal properties [15,16,17], improving insulation properties [18,19], improving electrical resistivity [20], and so on.

From a functional perspective, aggregates can serve as carriers for various materials. Some studies have investigated PCM impregnation. These studies usually aim to enhance the energy efficiency of buildings and infrastructure through energy storage. Sani et al. [21] developed and investigated the performance of PCM-impregnated lightweight aggregate. The process was complex, involving impregnation under vacuum conditions, coating with epoxy, and applying graphite on the surface of the coated aggregate. The developed aggregate was not applied to concrete but its performance was assessed in the raw state. Sani et al. [21] showed there was no leakage after thermal stress cycling and improved thermal storage performance. Using a similar method, Yoo et al. [22] developed an energy storage fine aggregate using zeolite. The production process used was the same as that of Sani et al. [21]; however, the materials were different. Sani et al. [21] used the PCMs of palm oil, coconut oil, and butter. The PCM used in the study of Yoo et al. [22] was paraffin wax. Zeolite is a kind of lightweight aggregate. The zeolite matrix contains many pores, a characteristic that facilitates PCM impregnation. With regard to the development of ESA, a study is reported by Kim et al. [23]. Yoo et al. [22] developed a fine aggregate for energy saving, while Kim et al. [23] developed an ESSBA. As is well known, aggregates occupy the greatest volume of concrete. Therefore, the aggregate substitution ratio of ESA is increased, and the energy storage performance is increased as well.

Most energy storage studies have focused on using PCMs, seeking to evaluate the energy storage performance. In addition, Kim et al. [23] performed a mock-up test comparing 100% ESSBA concrete compared to normal concrete. The results indicated a reduction in room temperature by approximately 4.5 °C using 100% ESSBA concrete. In the studies referred to energy storage performance was only demonstrated experimentally. There are also numerical analysis studies on PCM-containing concrete. However, these numerical studies have primarily simulated PCM behavior in concrete using predefined numerical properties [24,25,26] or models fitted to experimental data [27]. In addition, most PCM-based concrete studies roughly estimate economic benefits when using PCM-based concrete. It is necessary to undertake more precise and quantitative exploration of the economic/engineering benefits of PCM utilization, moving beyond approximate assessments. In simulations using time series data, there are two kinds of simulations that can be performed. The first is statistical simulation. An example of this approach is that of Woo et al. [28] who performed a Bayesian statistical simulation to assess the corrosion state of rebars. The Bayesian simulation indicated that the corrosion state could be indexed using statistical values, and a probability-based prediction of corrosion level could be used. The Bayesian indexing method produced quite accurate results, and Woo et al. [28] suggested that this process can be applied to real-time monitoring systems. It is possible to simulate time series data in a statistical manner and other Bayesian simulations can be found in various studies [17,29,30]. The principles of Bayesian statistics underpin machine learning and neural network processes. In consequence, the accuracy of Bayesian simulation results is quite high when the parameters are applied properly. However, a drawback of the Bayesian approach is that determining the posterior and parameter updates is difficult. As data size increases, the computation time also grows exponentially, impacting the efficiency of Bayesian estimation. Additionally, the characteristics of real-time monitoring systems result in a continuous accumulation of data. Consequently, there is a concern that the monitoring performance may also sharply decline. Furthermore, with the rapid proliferation of artificial intelligence following AlphaGo, the application of AI has become increasingly focused on time series data prediction in the construction industry. Examples of artificial intelligence utilization in the construction industry can be readily found. Jiang et al. [31] performed a prediction simulation on the thermal insulation temperature behavior of concrete using a modified RNN. Their results showed that the modified RNN predicted the thermal behavior of the concrete accurately, and the original RNN also predicted the behavior well. In another case, Khan et al. [32] carried out a compressive analysis to predict the strength of concrete considering various material parameters, such as the cement and aggregates used, and the curing time, using an optimized artificial neural network. The predicted compressive strength was almost the same as that observed experimentally and the reported R^2^ value was higher than 0.95, indicating that the results of Khan et al. [32] have high reliability. In addition to these cases, the use of artificial neural networks, such as LSTM [33,34,35] and convolution neural networks [36,37,38], in concrete studies has already demonstrated several times that the accuracy of prediction is high.

This study aimed to simulate and investigate energy consumption in concrete structures using mock-up test data. Therefore, it was necessary to add a function for retrieving signals after prediction. Hence, this study focused on the function of an SNN. SNNs are a kind of unsupervised learning and currently represent the next-generation of neural networks compared to CNNs [39]. A remarkable feature is that an SNN can derive binary signals based on a threshold. This makes SNNs light and fast in terms of computing [39]. However, there are distinct disadvantages as well. The accuracy of an SNN can be lower than for conventional neural networks, such as RNNs and LSTM, and more importantly, the inability to clearly define the backward process makes training the data challenging [39]. Thus, this study reports the design of an smRNN that is used in simulations that combines the definitive performance of an RNN in terms of learning and accuracy with the spiking mechanism of an SNN.

To simulate the power consumption between NC and EC100, data were used from the study of Kim et al. [23]. Because Kim et al. [23] installed the mock-up test specimens outside during the summer season in Korea and collected data for 1 month, specific periods were chosen in this study. Based on the experimental data, a simulation of the power consumption of the air conditioner was performed. This study offers insights into the combined utilization of smRNN and Bayesian thresholding to predict air conditioner operation times in concrete structures.

## 2. Materials and Methods

To perform the statistical simulation, a number of procedures were required. The process was quite complex; therefore, the entire process is presented in Figure 1 for ease of understanding.

This study adopted a more practical approach to power consumption than the studies introduced previously; however, it could not account for all the influences. Thus, some assumptions were established. The details are addressed in Section 2.4.

### 2.1. Data Collection

The study undertaken by Kim et al. [23] was performed from 28 July 2018 to 31 August 2018 in Seoul, Korea. Over the period of one month, a massive amount of data were collected, with readings taken every 10 min. Usually, Bayesian inference performs best when using massive datasets [28,30]. In this study, data were collected from 28 July 2018 to 14 August 2018. This period was chosen because it was the hottest summer since 1973 and coincided with a heat dome issue in Korea [40]. Hence, it was thought that the chosen period was suitable for analyzing/investigating the ESSBA concrete. The dimensions of the specimens and the experimental details are presented in Figure 2, and the composition of specimens is indicated in Table 1. The tested mixtures totaled 11 cases in the study of Kim et al. [23], but only two cases were applied to the mock-up test: NC and 100% substituted EC100.

Additional information on ESSBA is provided to describe the production process involved. First, a vacuum desiccator was prepared to impregnate the slag coarse aggregate with paraffin, which has a melting point of 47 °C. Next, paraffin blocks were placed into the desiccator, and the desiccator containing the paraffin blocks was placed in the heating chamber to fully melt the paraffin. After melting the paraffin blocks, the slag aggregate was placed into the paraffin liquid and the vacuum pump was operated to create an internal desiccator vacuum. In this state, the paraffin liquid was forced to fill the pores of the slag aggregate. This state was maintained for three days, and then the paraffin-soaked slag aggregate was removed from the desiccator. Before the surface cooled down, SiC was sprayed onto the surface of the aggregate. The process of making SiC-coated ESSBA was then fully completed. The process of producing the SiC-coated ESSBA is presented in Figure 3.

### 2.2. Discomfort Index

To assess something, various kinds of inputs are helpful to ensure precise examination. However, various types of inputs increase the columns of data structures, and it makes the computing times longer. Thus, it is best to use a representative index that includes many kinds of parameters because using this approach can reduce the columns of data structures. The most representative composite index commonly used is the DI. In addition, the DI is the most important indicator used as part of the statistical approach applied in this study. The DI considers the ambient temperature and the relative humidity. It represents human feelings of discomfort numerically. The DI is expressed according to Equation (1) [41].
(1)$$DI=T_{\;}-0.55(1-0.01RH)(T_{\;}-14.5)$$

The data used in this study comprised the indoor temperature of normal concrete TNC, the indoor temperature of ESSBA concrete TESSBA, the RH, and the outside air temperature, Ta. Data were collected every 10 min from 28 July 2018 at 12:10 to 14 August 2018 at 17:30. Based on the data obtained, each DI in NC (DI_NC_), EC100 (DI_ESSBA_), and the air (DI_a_) were calculated using Equation (1). The study focused on the difference between each DI. The core DI was DI_a_ and the supplementaries were DI_NC_ and DI_ESSBA_. The details of the DI differences are represented in Equations (2) and (3).
(2)$$DI_{ao}={DI}_{a}-DI_{NC}$$
(3)$$DI_{ae}=DI_{a}-DI_{ESSBA}$$

DI_NC_ and DI_ESSBA_ tended to be smaller than DI_a_. Hence, the larger values from DI_ao_ and DI_ae_ meant that the internal DI was getting smaller. This can be considered as a positive signal. However, the DI_ao_ and DI_ae_ values were smaller than 0, which means that the indoor DI was significantly higher than DI_a_. Based on DI_ao_ and DI_ae_, a Bayesian statistical simulation for estimating the power consumption by the air conditioner of a specific space was performed. To perform the simulation effectively, specific assumptions were needed. The assumptions are addressed in Section 2.4.

### 2.3. BT through the Baye’s Rule

It is good to use a representative index that includes many parameters in one output to avoid a complex data structure. Thus, DI_ao_ and DI_ae_ were introduced in the previous section. Using these indices, a BT could be calculated. Woo et al. [28] used only a voltage signal but calculated a BT following Baye’s rule of Equation (4) [42].
(4)$$p(\alpha |\beta )=\frac{p\left(\beta |\alpha \right)p(\alpha )}{p(\beta )}$$

In this state, the most challenging problem is the size of the dataset. According to Baye’s rules, the likelihood function p(β|α) = $\prod_{i=1}^{n}\;$p(βi|α), p(β|α) approaches zero when the dataset becomes bigger. The dataset used in this study was large enough to cause the p(β|α) value to approach 0; thus, the calculation of posterior values after finding the posterior mean and variance would be advantageous for thresholding. Considering the characteristics of likelihood derivation for the posterior function, this study focused on using a normal distribution as in Equation (5). The reason for choosing a normal distribution is discussed in Section 3.1.
(5)$$f(x)=\frac{exp\left(-\frac{1}{2}{\left(\frac{x-\mu }{\sigma }\right)}^{2}\right)}{\sqrt{2\pi \sigma^{2}}}$$

In this state, let $\sigma^{2}$ be known and μ be unknown, and let $\sigma_{0}^{2}$ and $\mu_{0}$ represent the variance and mean in the prior state. Then, the prior and the likelihood can be expressed as in Equations (6) and (7) [42].
(6)$$f(\mu )=\frac{exp\left(-\frac{{\left(\mu -\mu_{0}\right)}^{2}}{2\sigma_{0}^{2}}\right)}{{\left(2\pi \sigma_{0}^{2}\right)}^{1/2}}$$
(7)$$f(x|\mu )=\prod_{i=1}^{n}f\left(x_{i}|\mu \right)=\frac{exp\left(-\frac{\sum_{i=1}^{n}{\left(x_{i}-\mu \right)}^{2}}{2\sigma^{2}}\right)}{{\left(2\pi \sigma^{2}\right)}^{n/2}}$$

According to Equation (4), the marginals are usually considered to be constant; therefore, it can be expressed as p(α|β) ∝ p(β|α)p(α). In this state, the overall process of finding the posterior mean and the variance is as follows: With the proportional relationship before, the roughly expressed formation is as in Equations (8) and (9) [42].
(8)$$f(x|\mu )f(\mu )=\frac{exp\left(-\frac{\sum_{i=1}^{n}{\left(x_{i}-\mu \right)}^{2}}{2\sigma^{2}}\right)}{{\left(2\pi \sigma^{2}\right)}^{n/2}}\frac{exp\left(-\frac{{\left(\mu -\mu_{0}\right)}^{2}}{2\sigma_{0}^{2}}\right)}{{\left(2\pi \sigma_{0}^{2}\right)}^{1/2}}$$
(9)$$f(x|\mu )f(\mu )=\frac{1}{{\left(2\pi \sigma^{2}\right)}^{n/2}}\frac{1}{{\left(2\pi \sigma_{0}^{2}\right)}^{1/2}}exp\left(-\frac{1}{2}\left(\frac{\sum_{i=1}^{n}{\left(x_{i}-\mu \right)}^{2}}{\sigma^{2}}+\frac{{\left(\mu -\mu_{0}\right)}^{2}}{\sigma_{0}^{2}}\right)\right)$$

The terms outside of the exponential function are constants. Thus, arranging the details can focus on the terms inside of the exponential function, and let the inside terms be represented as a. Then, the arranging process can be simplified as Equation (10) and further processes are as Equations (11)–(14) [42]. Here, the term $\sum_{i=1}^{n}{\left(x_{i}-\mu \right)}^{2}$ can be expressed in more detail that $\sum_{i=1}^{n}{\left(x_{i}-\bar{x}+\bar{x}-\mu \right)}^{2}$ is the sample mean of $\bar{x}=\frac{\sum_{i=1}^{n}x_{i}}{n}$. In addition, $\sum_{i=1}^{n}{\left(x_{i}-\bar{x}+\bar{x}-\mu \right)}^{2}$ is able to be simplified to $\sum_{i=1}^{n}{\left(x_{i}-\bar{x}\right)}^{2}+n{\left(\bar{x}-\mu \right)}^{2}$ [42].
(10)$$a=\frac{\sum_{i=1}^{n}{\left(x_{i}-\mu \right)}^{2}}{2\sigma^{2}}+\frac{{\left(\mu -\mu_{0}\right)}^{2}}{2\sigma_{0}^{2}}$$
(11)$$a=\frac{\sum_{i=1}^{n}{\left(x_{i}-\bar{x}\right)}^{2}}{{2\sigma }^{2}}+\frac{n{\left(\bar{x}-\mu \right)}^{2}}{{2\sigma }^{2}}+\frac{{\left(\mu -\mu_{0}\right)}^{2}}{{2\sigma }_{0}^{2}}\propto \frac{n{\left(\bar{x}-\mu \right)}^{2}}{{2\sigma }^{2}}+\frac{{\left(\mu -\mu_{0}\right)}^{2}}{{2\sigma }_{0}^{2}}$$
(12)$$\propto \frac{n{\bar{x}}^{2}-2n\bar{x}\mu +n\mu^{2}}{{2\sigma }^{2}}+\frac{\mu^{2}-2\mu \mu_{0}+\mu_{0}^{2}}{2\sigma_{0}^{2}}$$
(13)$$\propto \frac{\mu^{2}}{2}\left(\frac{n}{\sigma^{2}}+\frac{1}{\sigma_{0}^{2}}\right)-\mu \left(\frac{n\bar{x}}{\sigma^{2}}+\frac{\mu_{0}}{\sigma_{0}^{2}}\right)+constant$$
(14)$$\propto \left(\frac{n}{\sigma^{2}}+\frac{1}{\sigma_{0}^{2}}\right)\left(\frac{\mu^{2}}{2}\right)-\mu \left(\frac{\frac{n\bar{x}}{\sigma^{2}}+\frac{\mu_{0}}{\sigma_{0}^{2}}}{\frac{n}{\sigma^{2}}+\frac{1}{\sigma_{0}^{2}}}\right)$$

Let the posterior follow the state of $N(\mu_{p},\sigma_{p}^{2})$, then the exponential term is $exp\left(-\frac{{\left(\mu -\mu_{p}\right)}^{2}}{2\sigma_{p}^{2}}\right)$. Therefore, $\mu_{p}$ = $\frac{\frac{n\bar{x}}{\sigma^{2}}+\frac{\mu_{0}}{\sigma_{0}^{2}}}{\frac{n}{\sigma^{2}}+\frac{1}{\sigma_{0}^{2}}}$ and $\sigma_{p}^{2}$ = ${\left(\frac{n}{\sigma^{2}}+\frac{1}{\sigma_{0}^{2}}\right)}^{-1}$. Using this posterior, a BT in each case can be calculated.

### 2.4. smRNN

The RNN has a simple hidden layer structure, but the prediction performance shows quite high accuracy. With the RNN it is easy to train the data, and the data structure of this study is quite simple. In addition, unlike LSTM which has a complex structure of the hidden layer, the hidden layer of the RNN is not complex. Thus, adding some functions is not difficult. In addition, an SNN is a very light neural network because an SNN imitates the data transmission process of neurons [39] as previously mentioned and shown in Figure 4.

Since the SNN process is simple, this study considered the main process to be the spiking part. Additionally, the spiking part was added in the hidden layer of the RNN process; therefore, it is referred to as smRNN in this study. The difference between a conventional RNN and smRNN is illustrated in Figure 5.

In the conventional RNN shown in Figure 5a, the red box location is empty. In contrast, the spiking part was added to the red box location as shown in Figure 5b. Hence, two kinds of data could be obtained from the smRNN—predicted outputs and spiked outputs. With the predicted outputs of T_NC_, T_ESSBA_, DI_ae_, and DI_ao_, the simulation obtained the spiked signals and predicted the energy consumption times of the air conditioner. The operation details of smRNN are listed in Table 2 and Table 3.

In Table 3, there are the well-known RNN parameters; Sherstinsky (2020) [43] arranged the process calculation of the traditional RNN system. The most important thing is the module-added part. The newly added part is highlighted with a dot-dash line in Table 3. The process is simple and not demanding; therefore, the training time did not increase. Thus, in this state, smooth data training could be performed. Additionally, the calculation of MSE and R^2^ is as in Equations (15) and (16).
(15)$$MSE=\frac{\sum {\left(\hat{y}-y\right)}^{2}}{n}$$
(16)$$R^{2}=1-\frac{\sum {\left(y-\hat{y}\right)}^{2}}{\sum {\left(y-\bar{y}\right)}^{2}}$$
where y are the observed data, $\hat{y}$ is the estimated output, $\bar{y}$ is the target average, and *n* is the amount of data. Considering the purpose of this study, since it is more meaningful to perform the simulation using the predicted data, prediction data were used instead of raw data. An example of future applications would be predicting the energy consumption for the next year or month in weather service centers.

### 2.5. Assumptions

First, the study aimed to estimate house-scale power consumption. There was a small space inside the specimens; therefore, the room temperature was quite high. Room temperature is usually higher in small rooms but can decrease as the size of the space increases. For example, consider two houses in the same climate without air conditioning systems: one is a large house, and the other is a small house. Usually, a large house has more shadow zones, which make the rooms cooler than those in a small house. Therefore, large indoor spaces traditionally have lower temperatures than small spaces [44]. In addition, small spaces can be highly affected by the radiation heat from the concrete walls. However, large spaces are less affected by the radiation heat than small spaces, as shown in Figure 6.

According to Figure 6, the red box area of overlapped radiation heat clearly shows higher temperatures than without a radiation heat space. However, assuming the space increases like the blue box area of Figure 6, it can create areas unaffected by the radiation heat from concrete walls. In this case, the room temperature would be lower than the red box case of Figure 6. The experiment undertaken by Kim et al. [23] is like the red box case of Figure 6; therefore, there is a need to correct the indoor temperatures. Including the temperature correction, the assumptions of this study are as follows:The indoor temperatures of the specimens should be subtracted from the original data by 5 °C considering the space size effect [44].The RH values are the same in air, indoor of NC, and EC100.The air conditioner is turned on when the room temperature is higher than 28 °C (temperature domination—red area).The air conditioner is turned on when DI_ao_ or DI_ae_ are downward of the baseline (index domination—condition—blue area).The air conditioner is turned on when both conditions 2 and 3 are satisfied at the same time (complex condition—green area).The air conditioner is turned off when DI_ao_ or DI_ae_ are above the baseline and the indoor temperature is below 28 °C (these two conditions should be satisfied at the same time).

In the introduced assumptions, there is reference to the ‘baseline’. The meaning of baseline is the connection line of BT and DI at the crossed point as shown in Figure 7.

The detailed results are considered in the following section, with the value of the baseline being twice the y-axis value. The reason that the crossed points were chosen was that there were some patterns to judge the simulated effects of turning the air conditioner on/off. The details are presented in Section 3. Based on the established assumptions, total power consumption estimation calculations were performed.

## 3. Results and Discussion

### 3.1. BT of the DIs

In general, the normal distribution is frequently used because it is easy to apply to simulations involving the use of trends and data [17]. However, there are many probability density functions that fit well with particular measured/prepared data, such as the Cauchy distribution and the gamma distribution. Therefore, it is essential to check what function fits the data well.

Before checking the fittings, the calculated DIs should be discussed. Figure 8 indicates the trends for DI_a_, DI_NC_, and DI_ESSBA_. As can be seen in Figure 8, DI_a_ usually showed higher values than DI_NC_ and DI_ESSBA_. This is because the indoor temperatures of NC and EC100 were subtracted by 5 °C and these subtracted temperatures largely affected the results of DI_NC_ and DI_ESSBA_. In addition, the trends for DI_a_ and DI_NC_ were generally observed in the DI values for both indoors and outdoors [45]. Thus, the observed trends were found to be general.

According to Figure 9, one thing becomes clear. The subtracted conditions of DI_ao_ and DI_ae_ showing positive values indicate that indoor conditions are more pleasant than outside. Furthermore, for the energy storage performance of ESSBA, it is observed in Figure 9 that the blue line has a lower position than the DI of air and NC for all periods. Therefore, it is worth considering the difference between DI values based on DI_a_; Figure 9 indicates the values of DI_ao_ and DI_ae_.

In most of the periods, DI_ae_ was above 0; however, for around 25% of periods DI_ao_ was below 0. In short, DI_ae_ showed that the indoor condition of EC100 was better than the indoor condition of NC. From the results of Figure 9, roughly, it can be expected that EC100 would show less consumption of electrical power for cooling the indoor space than NC. 

However, power consumption simulation could not be performed in this state; therefore, BT was needed for the simulation. Hence, finding the best-fitted probability function was performed for DI_ao_ and DI_ae_. To find the best probability function, histogram fitting, PP fitting, and QQ fitting were performed. These three methods are usually applied to obtain an appropriate probability function [30,46]. In the fitting work, normal, logistic, and Cauchy distributions were investigated. Figure 10 presents the histogram fitting.

These three cases of the distributions showed well-fitted conditions. Considering the skewness and kurtosis, the Cauchy distribution may appear to be the best; however, the rest of the distributions showed high-quality fitting as well. Thus, more data is required in order to determine the best fit of the distributions in DI_ao_ and DI_ae_. Hence, PP and QQ fitting were performed and are shown in Figure 11.

Figure 11a for DI_ae_ shows that all the distributions showed the same flow in PP fitting, but the details are different. The Cauchy distribution, as shown in Figure 11a, showed a greater distance from the ideal line than the normal and logistic distributions. Furthermore, in the cases of the Cauchy and logistic distributions, these two distributions exhibit thicknesses in their lines. In short, this may reflect a kind of noise when performing the thresholding work [28]. It is better to choose smoother lines in PP plots. In addition, the yellow area in the a-box of Figure 11a shows that a normal distribution was the best in terms of following the trend with the ideal line. This same trend is found in Figure 11b. In the QQ plots of Figure 11b, all the lines have the same width; however, it is clear that the normal distribution shows the best fit in the QQ of DI_ae_. In addition, a normal distribution shows complete accordance with the ideal lines for short duration in box b of Figure 11b. Thus, a normal distribution is appropriate for DI_ae_ indexing.

In the case of DI_ao_, determining the distribution was relatively more difficult than for DI_ae_. According to Figure 11c, the Cauchy distribution shows the best fit in the ranges of 0.4 to 0.6. However, in box c of Figure 11c, the normal distribution shows a better fit than for the Cauchy distribution. Hence, a cross-check with Figure 11d was carried out because Figure 11d shows a clear trend among the distributions. The left yellow box of Figure 11d shows that the normal distribution tends to follow the ideal line closer than the other distributions. On the other hand, the green box of Figure 11d shows the logistic and normal distributions behave almost without any differences. In the right yellow box in Figure 11d, the best-fitted distribution in DI_ao_ can be inferred by comparing with the results of Figure 11b,c. Overall, the normal distribution was the best probability density function in DI_ao_ as well. Based on the normal distribution, making the BT was performed. 

### 3.2. BT Results

Based on the derived posterior distribution, the BT lines showed the same value in the cases of DI_ao_ and DI_ae_. In Figure 12, each threshold value can be observed.

The temperature threshold is already introduced according to assumption 3. However, both NC and EC100 had the same threshold by the Bayes peak and the DI contrary to expectations. Since the Bayes peak could not reflect all the requirements of the assumptions, another criterion is essential to establish a basis [28]. For example, Woo et al. [28] used the Bayes peaks as a basis for assessing the corrosion state of the rebars. The base was the different peaks appearing along with the surface corrosion levels. Similarly, in this study, the DI curves serve as another criterion, with the value being 2 in all cases.

As can be seen in Figure 12, the Bayes peaks could not reflect all the assumptions about duration of this study. Due to the determined DI threshold, the simulation spiking in the smRNN was clearer than only with the Bayesian peaks. Therefore, all the conditions for simulating the air conditioning in this study were prepared.

### 3.3. Training and Test Results

The training and testing results are presented in Figure 13. To ensure usability of the fitted model, the accuracy of the test group must be high. In addition, the MSE loss during the training process should be close to 0, and the loss should be sufficiently stable after a certain number of epochs. This verification is discussed below.

The results are shown for T_NC_, T_ESSBA_, DI_ao_, and DI_ae_, respectively. The trained data showed close fitting with the raw data in all cases. In addition, the test data also showed close fitting. With simple structures of data, this can readily occur in RNN studies [47,48]. The close fitting and following exactly along with the raw data indicates why RNN is widely using in time series forecasting [47,48]. The smRNN training-testing performance confirms this in Figure 13 because the derived data show excellent fitting. However, it is difficult to determine how well the data fit quantitatively in this state. Thus, a comparison for the whole duration and raw data is given in Figure 14.

As expected, the R^2^ values in all cases were above 0.95. Above an R^2^ value of 0.8, it is considered that the predicted data fits well with the raw data and the prediction performance is quite high [17]. Furthermore, the lowest R^2^ value was 0.9708 in DI_ao_. Hence, using predicted data in this simulation can be performed with high reliability. In addition, in order to produce such a high R^2^ value, the MSE loss in the epoch process must be sufficiently low [49]. This is shown in Figure 15.

### 3.4. Air Conditioning Simulation Results

With the trained smRNN, the accurate predicted temperature and DI data are adequately derived. However, one additional type of data still remains to be considered—the spiked times. Since the duration is meaningful for the prediction part, it was derived together from the simulation analysis, and the duration can be accurately derived from the already trained smRNN. The results are presented in Figure 16.

Unlike Figure 12, the presented data are the predicted data from smRNN and confirm the other kinds of chart-like bars added in the graphs. The bar-shape charts are the simulated air conditioning times from the spiking module of the smRNN with the predicted temperature/DI data. The red spikes followed assumption 3. The spikes appeared when the predicted temperature data were above the temperature threshold of 28. The blue spikes followed assumption 4. The spikes appeared when the predicted DI values were lower than the DI threshold of 2. The green spikes represent assumption 5 of the complex condition.

An SNN is a kind of unsupervised learning but is very light. However, learning is difficult, and the accuracy of the results can be less than for conventional neural networks [39]. To overcome these shortcomings, learning was conducted through an RNN, and the final on/off simulation was derived through the spiking part of smRNN. Accurate results could be obtained as in Figure 16. From the on/off simulation, it can be seen that the spiked duration of NC and EC100 is clearly different. The NC simulation required that the air conditioner had to be turned on for more than 90% of the data measurement period. In contrast, in EC100, the operating period of the air conditioner was simulated to be significantly reduced. These results are in line with the experimental results of Kim et al. [23]. In addition, the difference in average indoor temperature of 3 °C in the simulation results is very similar to the results obtained by Kim et al. [23]. As the phase of PCM changed, the indoor temperature of the concrete structure was lowered in the process of absorbing and storing heat [22,23,50]. This suggests that the operating time of the air conditioner can be significantly reduced by a difference of 3 °C. The spiked duration details are presented in Table 4.

The total measured duration was 413.33 h. In the NC case, the total operating time was simulated by 390 h. It was 1.46 times more than for the EC100 case, and EC100 saved 32% of the operation time compared to NC. Thus, ESSBA achieved a reduction in energy consumption. Calculated simply by time, 123 h of energy consumption was saved, suggesting that this could lead to a positive chain effect on the burden of energy consumption at home and further carbon dioxide reduction in energy production.

## 4. Conclusions

This study sought to simulate energy consumption using smRNN. Two cases of concrete mock-up structures were introduced and data were obtained from a mock-up test. The key conclusions from this study are as follows:Through BT work, a clearer threshold could be derived. The probability peak compared to the data obtained from the posterior probability distribution did not enable the simulation required in this study. By providing a clear BT with DI, it was possible to obtain accurate spiking timing of smRNN.By implanting the spiking concept of SNN into RNN, predicted data and spiking timing could be obtained at the same time. During the training process, all losses were shown to be below 0.1, and the simulated data also showed an R^2^ value greater than 0.95. The meaning of smRNN made it possible to avoid secondary work on the prediction of the energy consumption duration. Faster and more accurate simulations were possible by obtaining results from the simulations at the same time.The use of PCM in the concrete showed remarkable energy consumption savings. According to the simulation undertaken in the study, the air conditioner operation time was reduced by 32% compared to the NC case. This implies that the utilization of PCM could be a key material to achieve energy consumption savings.The simulation process and methodology of this study can be used not only to predict energy efficiency but also by energy-related institutions to predict national-scale energy consumption during the summer.

## Figures and Tables

**Figure 1 materials-17-02108-f001:**
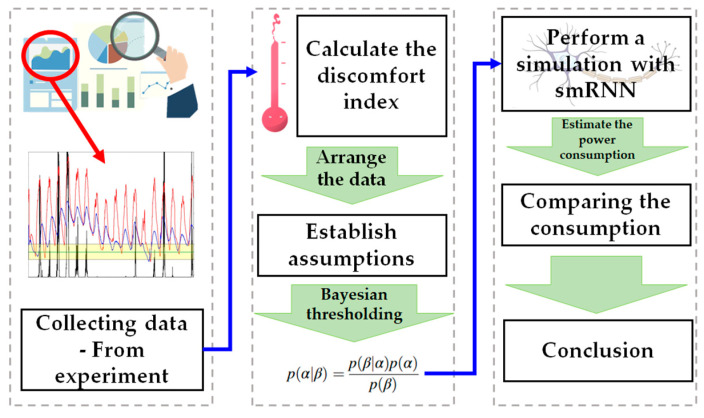
Research process.

**Figure 2 materials-17-02108-f002:**
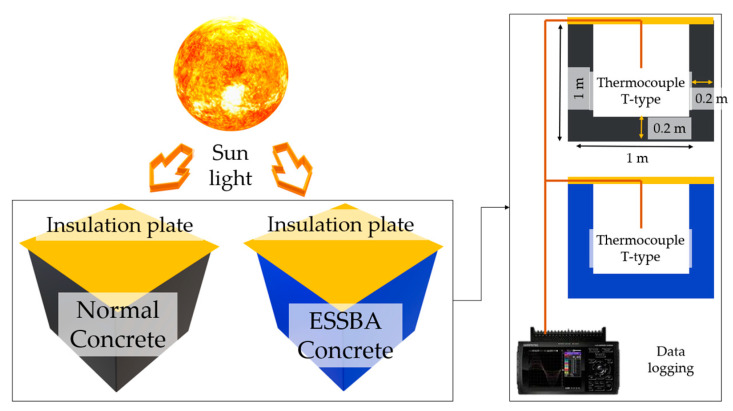
Details of the specimens and the experiment.

**Figure 3 materials-17-02108-f003:**
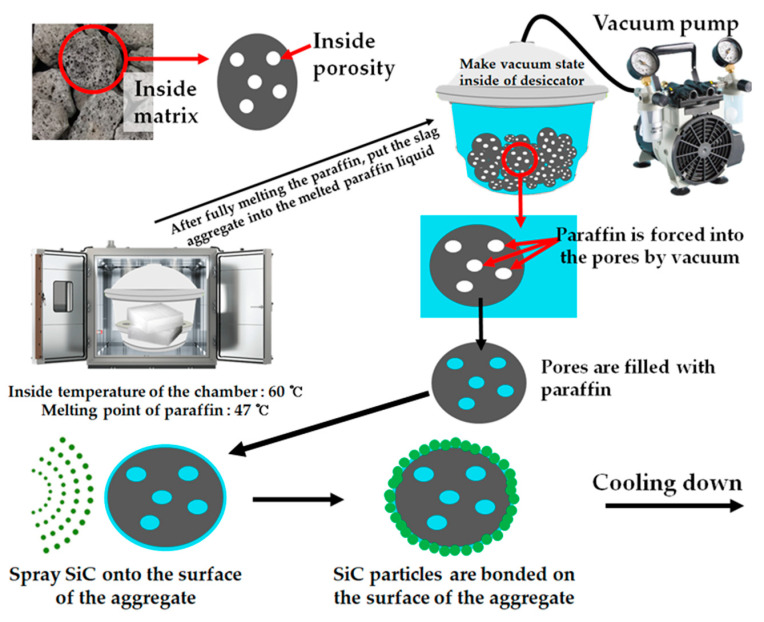
SiC-coated ESSBA making process.

**Figure 4 materials-17-02108-f004:**
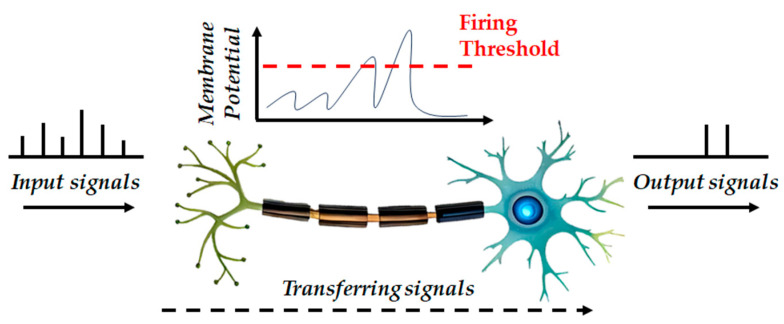
SNN process.

**Figure 5 materials-17-02108-f005:**
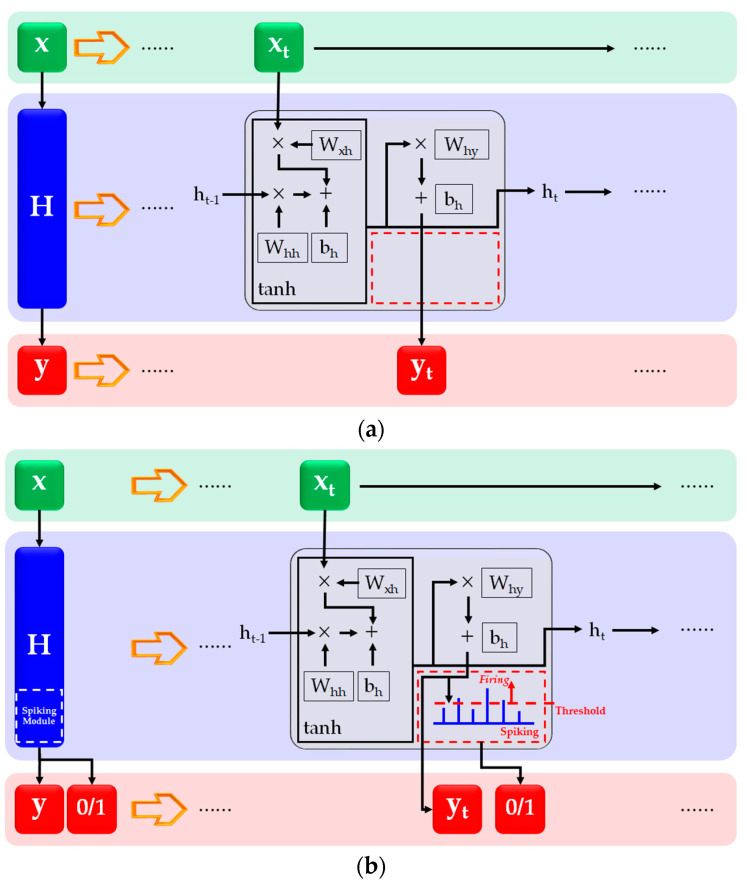
Difference between RNN and smRNN, (**a**) Conventional RNN structure, (**b**) smRNN structure.

**Figure 6 materials-17-02108-f006:**
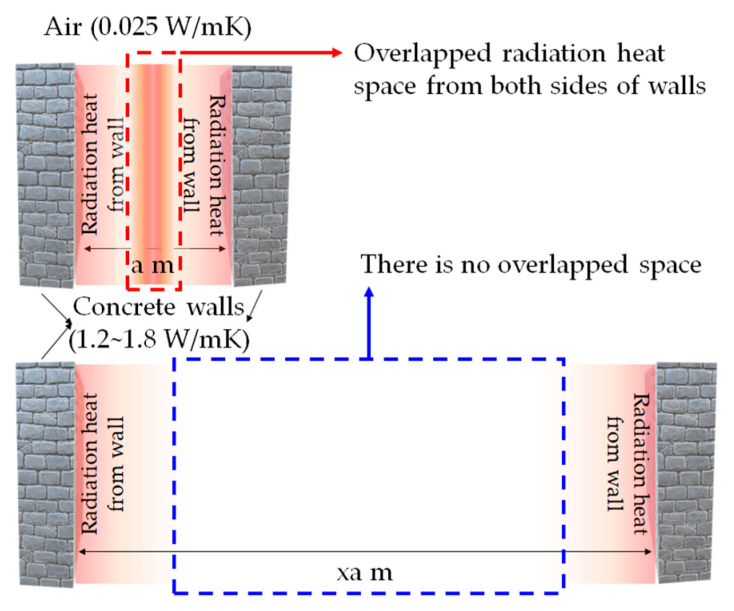
The effect of radiation heat from concrete walls on the indoor space.

**Figure 7 materials-17-02108-f007:**
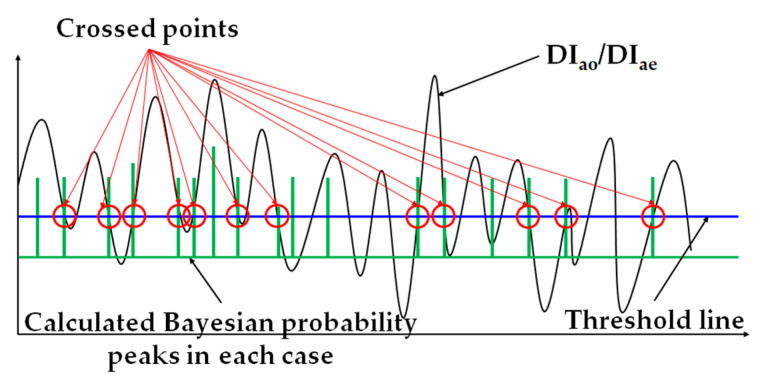
The concept of BT.

**Figure 8 materials-17-02108-f008:**
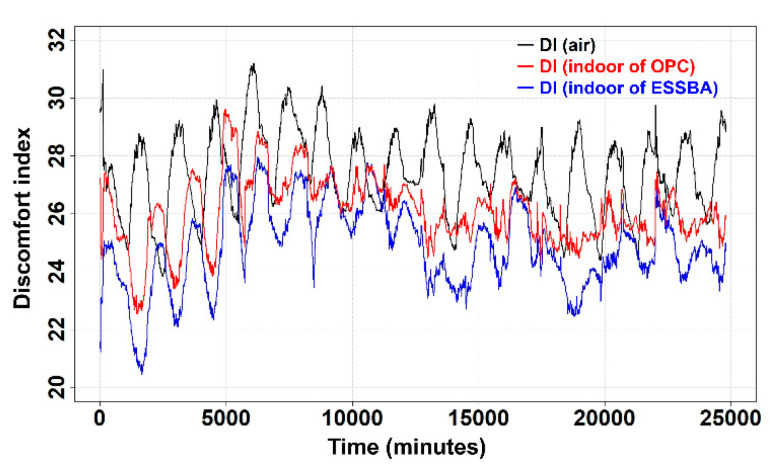
Calculated DIs of air, indoor of NC and ESSBA.

**Figure 9 materials-17-02108-f009:**
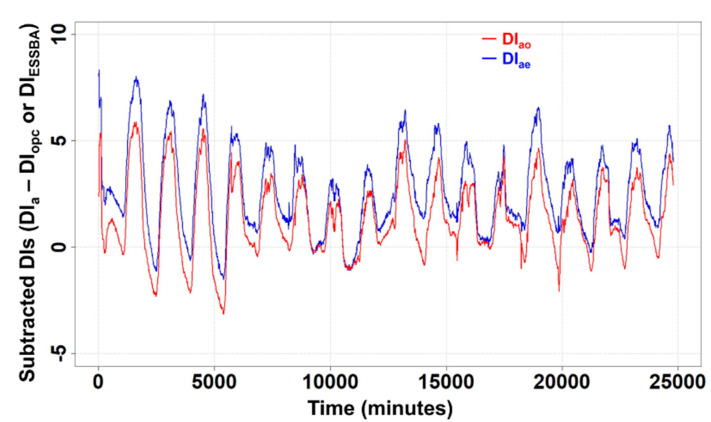
Calculated DI_ao_ and DI_ae_.

**Figure 10 materials-17-02108-f010:**
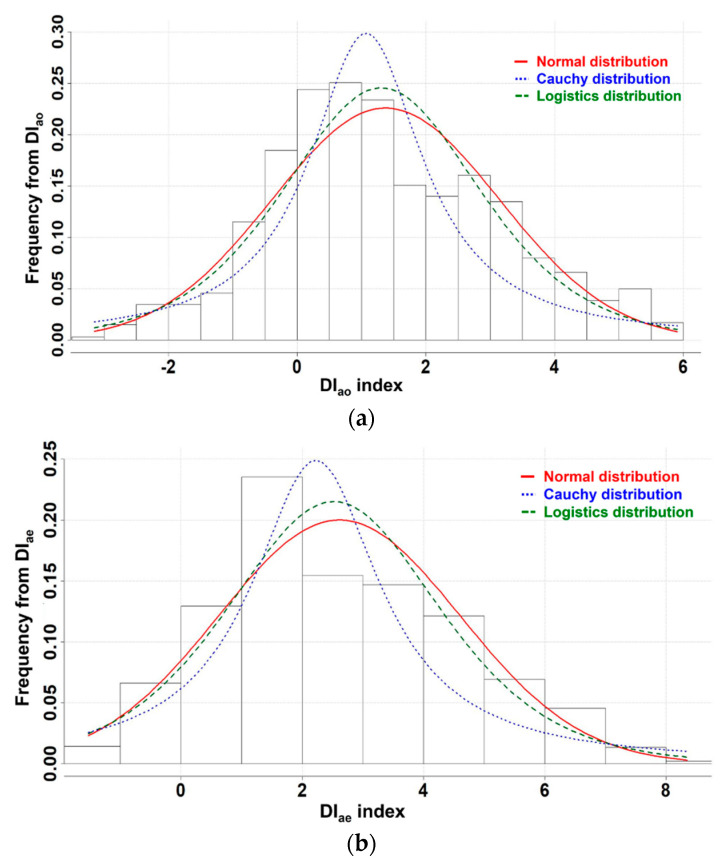
Histogram fitting results, (**a**) Histogram fitting of DI_ao_, (**b**) Histogram fitting of DI_ae_.

**Figure 11 materials-17-02108-f011:**
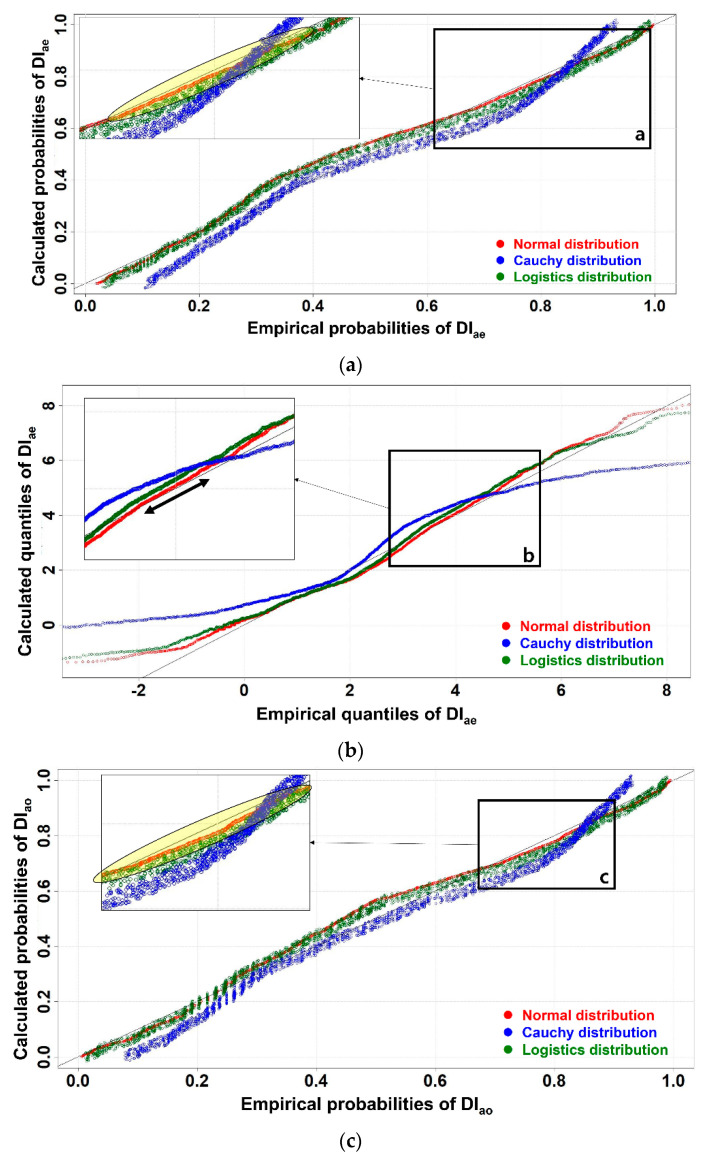
PP and QQ fitting in the cases of DI_ae_ and DI_ao_, (**a**) PP of DI_ae_, (**b**) QQ of DI_ae_, (**c**) PP of DI_ao_, (**d**) QQ of DI_ao_.

**Figure 12 materials-17-02108-f012:**
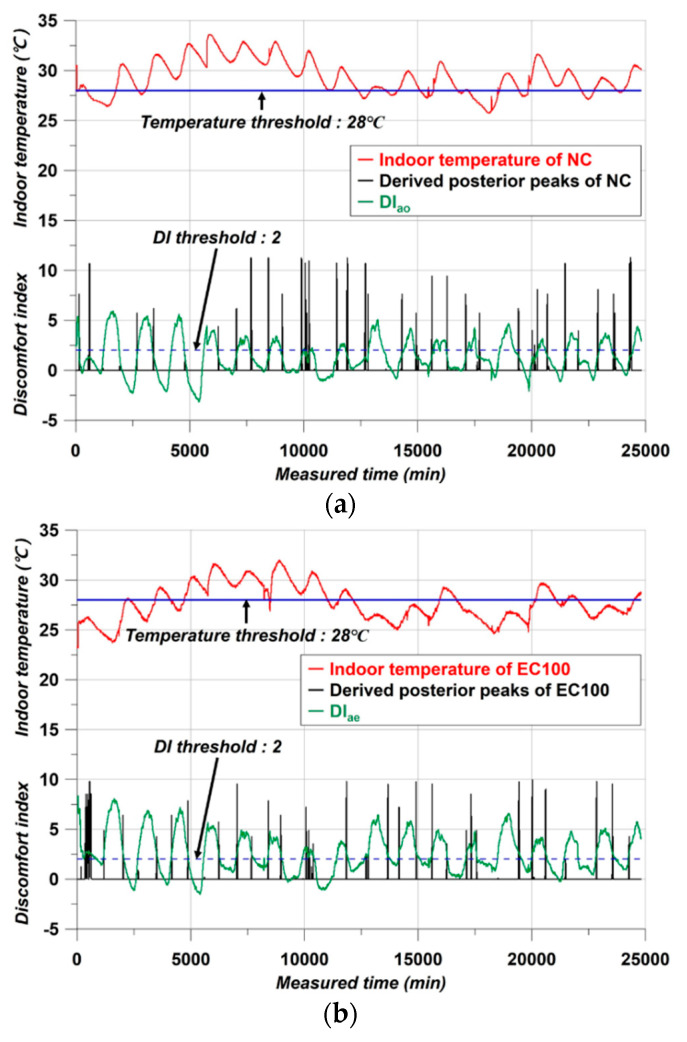
Thresholds, (**a**) NC case, (**b**) EC100 case.

**Figure 13 materials-17-02108-f013:**
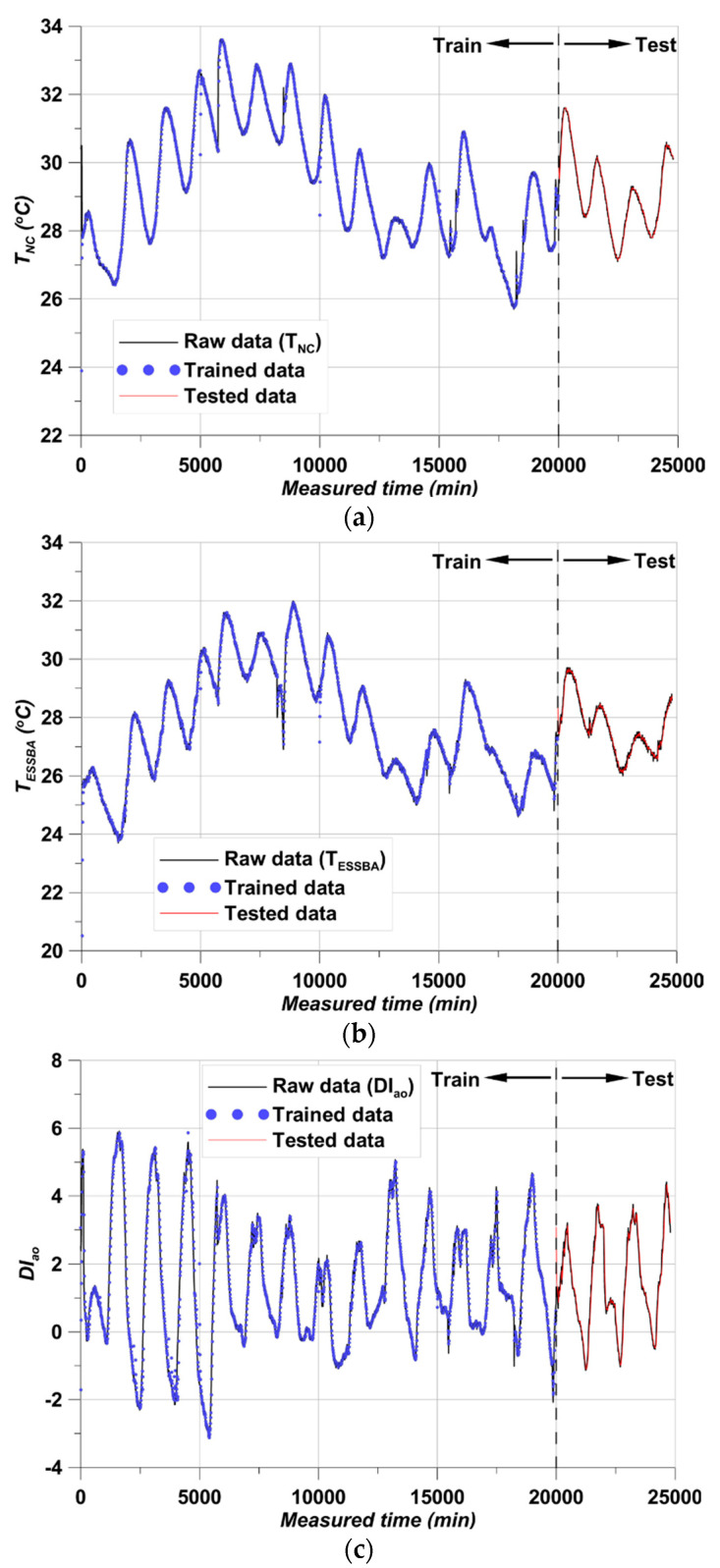
Train-test model test results, (**a**) T_NC_ results, (**b**) T_ESSBA_ results, (**c**) DI_ao_ results, (**d**) DI_ae_ results.

**Figure 14 materials-17-02108-f014:**
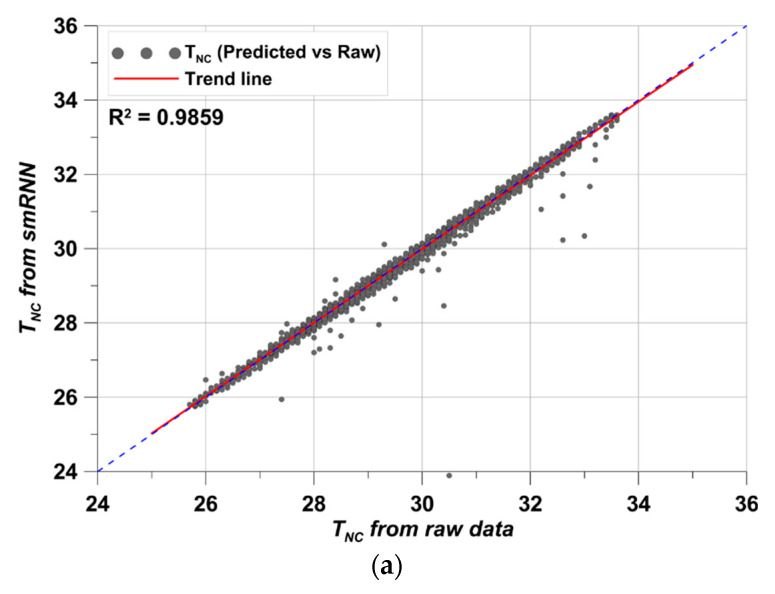
Data comparison, (**a**) T_NC_ results, (**b**) T_ESSBA_ results, (**c**) DI_ao_ results, (**d**) DI_ae_ results.

**Figure 15 materials-17-02108-f015:**
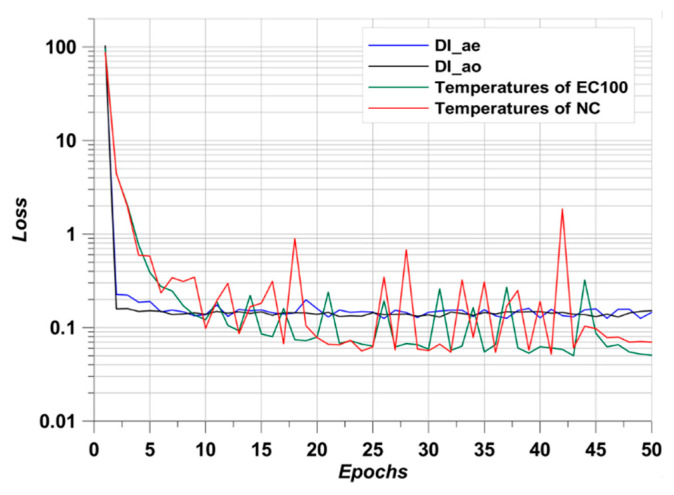
MSE loss along with epochs.

**Figure 16 materials-17-02108-f016:**
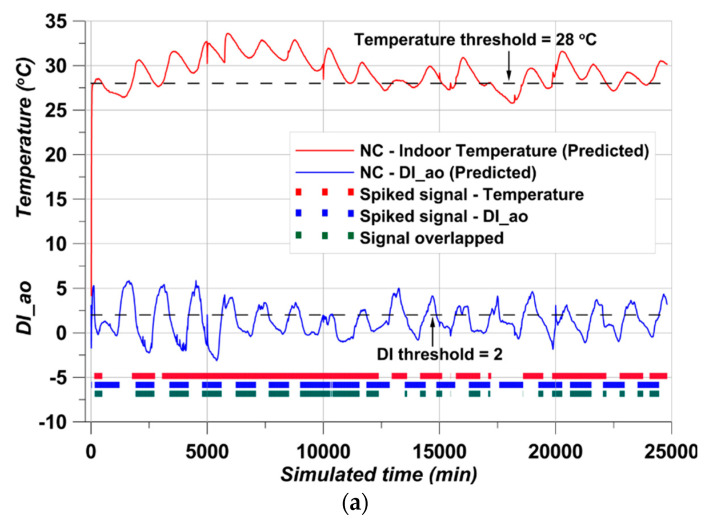
Simulation results, (**a**) The case of NC, (**b**) The case of EC100.

**Table 1 materials-17-02108-t001:** Mix properties of the specimens [23].

W/C (%)	S/a (%)	Unit: Kg, G_max_ = 25 mm						Specimen Name
		Water	Cement	Fine Aggregate	Coarse Aggregate	ESSBA	WR	
45	45.8	180	400	766	914	0	2.01	NC
					0	914		EC100

**Table 2 materials-17-02108-t002:** smRNN details.

**Used data**	t (total 24,800 min), T_NC_, T_ESSBA_, DI_ao_, DI_ae_
**Trainings**	90% of dataset
**Tests**	Rest 10% of dataset
**Thresholds**	T_NC_, T_ESSBA_: 28 °C DI_ao_, DI_ae_: Followed the results in Section 3.2
**Optimizer**	Gradient descent

**Table 3 materials-17-02108-t003:** Total simulation process with smRNN.

Air Conditioning Simulation Process	
**1**	**Load dataset**
**2**	**Divide the dataset** *Training and Testing*
**3**	**Initialize the parameters and thresholds** *W_xh_, W_hh_, W_hy_, b_h_, b_y_* *Learning rate = 0.001* *Hidden size = length(training set)* *Epochs = 50* *thr* ≥ *28 for temperature, *≤ *2 for DI* *Activation function (Forward) = tanh(x)* *Activation function (Backward) = dtanh(x)/dx*
**4**	**for** i **in** 1:epochs ## Training the model *h_ih_ = 0* ## Initialize the hidden state *spike = 0* ## Initialize the spike state **for** h **in** 1:hidden size ## Forward process *Calculate : h and y_train_out_ with x_train_[h]*
	## Backpropagation
	*Calculate : dy, db_y_, dW_hy_, dh, db_h_, dW_xh_, dW_hh_*
	*-----------------------------------------------------------------------------------------------* ## Get spikes *y_train-out_* $\overset{thr}{\to }$ *get 1 or 0 (1 = firing, 0 = non)* *-----------------------------------------------------------------------------------------------* *## Get loss* *MSE loss* *## Update the weights and biases (Optimizing)* *W_xh_, W_hh_, W_hy_, b_h_, b_y_*
	**end** **end**
**5**	**Test the trained model** *x_test_*$\overset{trained\;smRNN}{\to }$ *y_prediction_*
**6**	**Perform simulation if R^2^ value upper than 0.9 (Raw data vs. Predicted data)**

**Table 4 materials-17-02108-t004:** Spiked duration (operating time of the air conditioner).

Assumptions	Unit (hours)	
	NC	EC100
Temperature domination	307.667	168.333
Index domination	271.833	185.167
Complex condition	189.5	86.5
Total (except for overlapping duration)	390	267

## Data Availability

Data are contained within the article.

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
