# Peer review of "Machine Learning-Based Simulation of the Air Conditioner Operating Time in Concrete Structures with Bayesian Thresholding"

_materials, 2024, doi:10.3390/ma17092108_

Round 1

Reviewer 1 Report

Comments and Suggestions for Authors

Please find the comments referring to the article as an attachment.

Author Response

Respected Reviewer,

Thank you for reviewing our manuscript entitled Machine Learning based Simulation on the Air Conditioner Operating Time in Concrete Structures with Bayesian Thresholding for possible publication in the journal of “Materials”. We are thankful to you for your quick and valuable feedbacks to improve the quality of our manuscript for possible publication in the journal. We have revised the manuscript according to your comments and suggestions. The pointwise replies are given here in the response file.

Reviewer 2 Report

Comments and Suggestions for Authors

The article's topic is interesting and worth studying. Its theme and content recommend it for inclusion in the special issue State-of-the-Art Construction Materials and Technologies for Structural Health Monitoring of Infrastructures.

This paper outlines and interprets the authors’ perspective on using machine learning to simulate air conditioner operating times in concrete structures, emphasizing the role of Bayesian thresholding in this process.

The article has a satisfactory structure, but some parts and characteristics require improvement.

The abstract and introduction well position the performed research. The methodology and the results are presented satisfactorily.

The conclusions section is short and general and may be improved. Furthermore, implications for the obtained results and future research perspectives can be included.

Also, the quality of English can be improved. Try to avoid repeating words and phrases in the same paragraph (applicable to the entire manuscript) and reword the text in an academic style.

Overall, I found this work interesting and beneficial.

Comments on the Quality of English Language

The quality of English can be improved. Avoid repeating words and phrases in the same paragraph (applicable to the entire manuscript) and reword the text in an academic style.

Author Response

(The authors gave the same response as above.)

Reviewer 3 Report

Comments and Suggestions for Authors

Dear authors, there are several points for improvement in your manuscript.

1. In the text: "Warnings of the climatologist have been around for a long time" is not clear, it is urged to modify it.

2. They must include orthographic connectors before starting with the author explanations (for example, on lines 32 and 38).

3. The paragraphs that is on lines 60 to 68, 139 to 140, 156-157, 180-187, 205-207, 234-239, 299-300, and 359-361, are not clear, you should rewrite it. In addition, there are lines with verb conjugation in the past and present ¿?

4. Points 1 to 6 (paragraphs 287 to 298) should be written to avoid confusion. Including points 1 and 3 of the conclusions.

5. The text in Figure 2 and Figure 7 should be more precise and clear.

6. References have to be rewritten to meet the specifications of the journal editors.

Comments on the Quality of English Language

In lines 167 to 169 is an example of verb conjugation confusion.

Author Response

(The authors gave the same response as above.)

Round 2

Reviewer 1 Report

Comments and Suggestions for Authors

All comments of the reviewer have been included in the revised version of the article. I recommend publication of this article in its current form.

Author Response

(The authors gave the same response as above.)

Reviewer 3 Report

Comments and Suggestions for Authors

Dear authors,

Although his manuscript presents improvements, doubts still remain.

1. The paragraph located on lines 167-175 is confusing: they first begin with “he was prepared” and continue with “Next, put the parafin…”. The paragraph located on lines 268-273 likewise.

2. Point 2 of the conclusions is not understandable.

Author Response

(The authors gave the same response as above.)
